# Mo′ States Mo′ Problems:
# Emergency Stop Mechanisms from Observation

**Samuel Ainsworth**　　**Matt Barnes**　　**Siddhartha Srinivasa**

School of Computer Science and Engineering
University of Washington
`{skainswo,mbarnes,siddh}@cs.washington.edu`

## Abstract

In many environments, only a relatively small subset of the complete state space is necessary in order to accomplish a given task. We develop a simple technique using emergency stops (e-stops) to exploit this phenomenon. Using e-stops significantly improves sample complexity by reducing the amount of required exploration, while retaining a performance bound that efficiently trades off the rate of convergence with a small asymptotic sub-optimality gap. We analyze the regret behavior of e-stops and present empirical results in discrete and continuous settings demonstrating that our reset mechanism can provide order-of-magnitude speedups on top of existing reinforcement learning methods.

## 1   Introduction

In this paper, we consider the problem of determining when along a training roll-out feedback from the environment is no longer beneficial, and an intervention such as resetting the agent to the initial state distribution is warranted. We show that such interventions can naturally trade off a small sub-optimality gap for a dramatic decrease in sample complexity. In particular, we focus on the reinforcement learning setting in which the agent has access to a reward signal in addition to either (a) an expert supervisor triggering the e-stop mechanism in real-time or (b) expert state-only demonstrations used to "learn" an automatic e-stop trigger. Both settings fall into the same framework.

Evidence already suggests that using simple, manually-designed heuristic resets can dramatically improve training time. For example, the classic pole-balancing problem originally introduced in Widrow and Smith [25] prematurely terminates an episode and resets to an initial distribution whenever the pole exceeds some fixed angle off-vertical. More subtly, these manually designed reset rules are hard-coded into many popular OpenAI gym environments [7].

Some recent approaches have demonstrated empirical success learning when to intervene, either in the form of resetting, collecting expert feedback, or falling back to a safe policy [8, 16, 19, 14]. We specifically study reset mechanisms which are more natural for human operators to provide – in the form of large red buttons, for example – and thus perhaps less noisy than action or value feedback [5]. Further, we show how to build automatic reset mechanisms from state-only observations which are often widely available, e.g. in the form of videos [24].

The key idea of our method is to build a *support set* related to the expert's state-visitation probabilities, and to terminate the episode with a large penalty when the agent leaves this set, visualized in Fig. 1. This support set defines a modified MDP and can either be constructed implicitly via an expert supervisor triggering e-stops in real-time or constructed a priori based on observation-only roll-outs from an expert policy. As we will show, using a support set explicitly restricts exploration to a

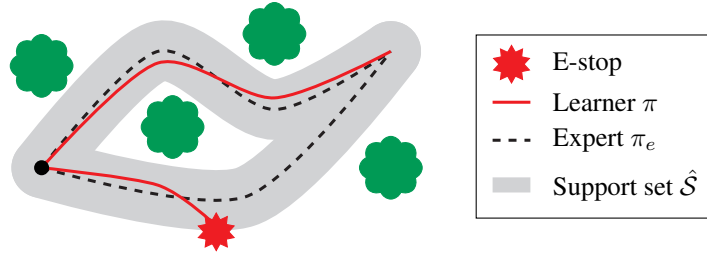

Figure 1: A robot is tasked with reaching a goal in a cluttered environment. Our method allows incorporating e-stop interventions into any reinforcement learning algorithm. The grey support set may either be implicit (from a supervisor) or, if available, explicitly constructed from demonstrations.

smaller state space while maintaining guarantees on the learner's performance. We emphasize that our technique for incorporating observations applies to *any* reinforcement learning algorithm in either continuous or discrete domains.

The contributions and organization of the remainder of the paper is as follows.

- We provide a general framework for incorporating arbitrary emergency stop (e-stop) interventions from a supervisor into any reinforcement learning algorithm using the notion of support sets in Section 4.

- We present methods and analysis for building support sets from observations in Section 5, allowing for the creation of automatic e-stop devices.

- In Section 6 we empirically demonstrate on benchmark discrete and continuous domains that our reset mechanism allows us to naturally trade off a small asymptotic sub-optimality gap for significantly improved convergence rates with any reinforcement learning method.

- Finally, in Section 7, we generalize the concept of support sets to a spectrum of set types and discuss their respective tradeoffs.

## 2 Related Work

The problem of learning when to intervene has been studied in several contexts and generally falls under the framework of safe reinforcement learning [10] or reducing expert feedback [16]. Richter and Roy [19] use an auto-encoder as an anomaly detector to determine when a high dimensional state is anomalous, and revert to a safe policy. Laskey et al. [16] use a one-class SVM as an anomaly detector, but instead for the purposes of reducing the amount of imitation learning feedback during DAGGER training [20]. Garcia and Fernández [9] perturb a baseline policy and request action feedback if the current state exceeds a minimum distance from any demonstration. Geramifard et al. [11] assume access to a function which indicates whether a state is safe, and determines the risk of the current state by Monte Carlo roll-outs. Similarly, "shielding" [3] uses a manually specified safety constraint and a coarse, conservative abstraction of the dynamics to prevent an agent from violating the safety constraint. Eysenbach et al. [8] learn a second "soft reset" policy (in addition to the standard "hard" reset) which prevents the agent from entering nearly non-reversible states and returns the agent to an initial state. Hard resets are required whenever the soft reset policy fails to terminate in a manually defined set of safe states $\mathcal{S}_{\text{reset}}$. Our method can be seen as learning $\mathcal{S}_{\text{reset}}$ from observation. Their method trades off hard resets for soft resets, whereas ours learns when to perform the hard resets.

The general problem of Learning from Demonstration (LfD) has been studied in a variety of contexts. In inverse reinforcement learning, Abbeel and Ng [1] assume access to state-only trajectory demonstrations and attempt to learn an unknown reward function. In imitation learning, Ross et al. [20] study the distribution mismatch problem of behavior cloning and propose DAGGER, which collects action feedback at states visited by the current policy. GAIL addresses the problem of imitating a set of fixed trajectories by minimizing the Jensen-Shannon divergence between the policies' average-state-action distributions [13]. This reduces to optimizing a GAN-style minimax objective with a reinforcement learning update (the generator) and a divergence estimator (the discriminator).

The setting most similar to ours is Reinforcement Learning with Expert Demonstrations (RLED), where we observe both the expert's states and actions in addition to a reward function. Abbeel and Ng [2] use state-action trajectory demonstrations to initialize a model-based RL algorithm, which eliminates the need for explicit exploration and can avoid visiting all of the state-action space. Smart and Kaelbling [22] bootstrap Q-values from expert state-action demonstrations. Maire and Bulitko [17] initialize any value-function based RL algorithm by using the shortest observed path from each state to the goal and generalize these results to unvisited states via a graph Laplacian. Nagabandi et al. [18] learn a model from state-action demonstrations and use model-predictive control to initialize a model-free RL agent via behavior cloning. It is possible to extend our method to RLED by constructing a support superset based on state-action pairs, as described in Section 7. Thus, our method and many RLED methods are complimentary. For example, DQfD [12] would allow pre-training the policy from the state-action demonstrations, whereas ours reduces exploration during the on-policy learning phase.

Most existing techniques for bootstrapping RL are not applicable to our setting because they require either (a) state-action observations, (b) online expert feedback, (c) solving a reinforcement learning problem in the original state space, incurring the same complexity as simply solving the original RL problem, or (d) provide no guarantees, even for the tabular setting. Further, since our method is equivalent to a one-time modification of the underlying MDP, it can be used to improve any existing reinforcement learning algorithm and may be combined with other bootstrapping methods.

## 3   Problem setup

Let $M = \langle \mathcal{S}, \mathcal{A}, P, R, H, \rho_0 \rangle$ be a finite horizon, episodic Markov decision process (MDP) defined by the tuple $M$, where $\mathcal{S}$ is a set of states, $\mathcal{A}$ is a set of actions, $P : \mathcal{S} \times \mathcal{A} \rightarrow \Delta(\mathcal{S})$ is the transition probability distribution and $\Delta(\cdot)$ is a probability simplex over some space, $R : \mathcal{S}^2 \times \mathcal{A} \rightarrow [0,1]$ is the reward function, $H$ is the time horizon and $\rho_0 \in \Delta(\mathcal{S})$ is the distribution of the initial state $s_0$. Let $\pi \in \Pi : \mathbb{N}_{1:H} \times \mathcal{S} \rightarrow \Delta(\mathcal{A})$ be our learner's policy and $\pi_e$ be the potentially sub-optimal expert policy (for now assume the realizability setting, $\pi_e \in \Pi$).

The state distribution of policy $\pi$ at time $t$ is defined recursively as

$$\rho_\pi^{t+1}(s) = \sum_{s_t, a_t} \rho_\pi^t(s_t)\pi(a_t|s_t,t)P(s_t, a_t, s) \qquad \rho_\pi^0(s) \equiv \rho_0(s). \tag{1}$$

The expected sum of rewards over a single episode is defined as

$$J(\pi) = \mathbb{E}_{s \sim \rho_\pi, a \sim \pi(s), s' \sim P(s,a)} R(s, a, s') \tag{2}$$

where $\rho_\pi$ denotes the average state distribution, $\rho_\pi = \frac{1}{H} \sum_{t=0}^{H-1} \rho_\pi^t$.

Our objective is to learn a policy $\pi$ which minimizes the notion of external regret over $K$ episodes. Let $T = KH$ denote the total number of time steps elapsed, $(r_1, \ldots, r_T)$ be the sequence of rewards generated by running algorithm $\mathfrak{A}$ in $M$ and $R_T = \sum_{t=1}^{T} r_t$ be the cumulative reward. Then the $T$-step expected regret of $\mathfrak{A}$ in $M$ compared to the expert is defined as

$$\text{Regret}_M^{\mathfrak{A}}(T) := \mathbb{E}_M^{\pi_e}[R_T] - \mathbb{E}_M^{\mathfrak{A}}[R_T]. \tag{3}$$

Typical regret bounds in the discrete setting are some polynomial of the relevant quantities $|\mathcal{S}|$, $|\mathcal{A}|$, $T$, and $H$. We assume we are given such an RL algorithm. Later, we assume access to either a supervisor who can provide e-stop feedback or a set of demonstration roll-outs $D = \{\tau^{(1)}, \ldots, \tau^{(n)}\}$ of an expert policy $\pi_e$ in $M$, and show how these can affect the regret bounds. In particular, we are interested in using $D$ to decrease the effective size of the state space $\mathcal{S}$, thereby reducing the amount of required exploration when learning in $M$.

## 4   Incorporating e-stop interventions

In the simplest setting, we have access to an external supervisor who provides minimal online feedback in the form of an e-stop device triggered whenever the agent visits states outside of some to-be-determined set $\hat{\mathcal{S}} \subseteq \mathcal{S}$. For example, if a mobile robot is navigating across a cluttered room as in Fig. 1, a reasonable policy will rarely collide into objects or navigate into other rooms which are

unrelated to the current task, and the supervisor may trigger the e-stop device if the robot exhibits either of those behaviors. The analysis for other support types (e.g. state-action, time-dependent, visitation count) are similar, and their trade-offs are discussed in Section 7.

## 4.1 The sample complexity and asymptotic sub-optimality trade-off

We argue that for many practical MDPs, $\rho_{\pi_e}$ is near-zero in much of the state space, and constructing an appropriate $\hat{\mathcal{S}}$ enables efficiently trading off asymptotic sub-optimality for potentially significantly improved convergence rate. Given some reinforcement learning algorithm $\mathfrak{A}$, we proceed by running $\mathfrak{A}$ on a newly constructed "e-stop" MDP $\widehat{M} = (\hat{\mathcal{S}}, \mathcal{A}, P_{\hat{\mathcal{S}}}, R_{\hat{\mathcal{S}}}, H, \rho_0)$. Intuitively, whenever the current policy leaves $\hat{\mathcal{S}}$, the e-stop prematurely terminates the current episode with no further reward (the maximum penalty). These new transition and reward functions are defined as

$$P_{\hat{\mathcal{S}}}(s_t, a_t, s_{t+1}) = \begin{cases} P(s_t, a_t, s_{t+1}), & \text{if } s' \in \hat{\mathcal{S}} \\ \sum_{s' \notin \hat{\mathcal{S}}} P(s_t, a_t, s'), & s_{t+1} = s_{\text{term}} \\ 0, & \text{else} \end{cases} \tag{4}$$

$$R_{\hat{\mathcal{S}}}(s_t, a_t, s_{t+1}) = \begin{cases} R(s_t, a_t, s_{t+1}), & \text{if } s_{t+1} \in \hat{\mathcal{S}} \\ 0, & \text{else} \end{cases}$$

where $s_{\text{term}}$ is an absorbing state with no reward. A similar idea was discussed for the imitation learning problem in [21].

The key trade-off we attempt to balance is between the *asymptotic sub-optimality* and *reinforcement learning regret*,

$$\text{Regret}(T) \leq \underbrace{\left\lceil \tfrac{T}{H} \right\rceil [J(\pi^*) - J(\hat{\pi}^*)]}_{\text{Asymptotic sub-optimality}} + \underbrace{\mathbb{E}_{\widehat{M}}^{\hat{\pi}^*}[R_T] - \mathbb{E}_{\widehat{M}}^{\mathfrak{A}}[R_T]}_{\text{Learning regret}}. \tag{5}$$

where $\pi^*$ and $\hat{\pi}^*$ are the optimal policies in $M$ and $\widehat{M}$, respectively (proof in Appendix A). The first term is due to the approximation error introduced when constructing $\widehat{M}$, and depends entirely on our choice of $\hat{\mathcal{S}}$. The second term is the familiar reinforcement learning regret, e.g. Azar et al. [4] recently proved an upper regret bound of $\sqrt{H|\mathcal{S}||\mathcal{A}|T} + H^2|\mathcal{S}|^2|\mathcal{A}|$. We refer the reader to Kakade et al. [15] for an overview of state-of-the-art regret bounds in episodic MDPs.

Our focus is primarily on the first term, which in turn decreases the learning regret of the second term via $|\hat{\mathcal{S}}|$ (typically quadratically). This forms the basis for the key performance trade-off. We introduce bounds for the first term in various conditions, which inform our proposed methods. By intelligently modifying $M$ through e-stop interventions, we can decrease the required exploration and allow for the early termination of uninformative, low-reward trajectories. Note that the reinforcement learning complexity of $\mathfrak{A}$ is now independent of $\mathcal{S}$, and instead dependent on $\hat{\mathcal{S}}$ according to the same polynomial factors. Depending on the MDP and expert policy, this set may be significantly smaller than the full set. In return, we pay a worst case asymptotic sub-optimality penalty.

An additional benefit of the e-stop MDP $\widehat{M}$ not captured in Eq. (5) is the ability to terminate trajectories upon entering state $s_{\text{term}}$. No learning is required for this state due to our perfect knowledge about its rewards and dynamics.

## 4.2 Perfect e-stops

To begin, consider an idealized setting, $\hat{\mathcal{S}} = \{s|h(s) > 0\}$ (or some superset thereof) where $h(s)$ is the probability $\pi_e$ visits state $s$ at any point during an episode. Then the modified MDP $\widehat{M}$ has an optimal policy which achieves at least the same reward as $\pi_e$ on the true MDP $M$.

**Theorem 4.1.** *Suppose $\widehat{M}$ is an e-stop variant of $M$ such that $\hat{\mathcal{S}} = \{s|h(s) > 0\}$ where $h(s)$ denotes the probability of hitting state $s$ in a roll-out of $\pi_e$. Let $\hat{\pi}^* = \arg\max_{\pi \in \Pi} J_{\widehat{M}}(\pi)$ be the optimal policy in $\widehat{M}$. Then $J(\hat{\pi}^*) \geq J(\pi_e)$.*

In other words, and not surprisingly, if the expert policy never visits a state, then we pay no penalty for removing it. (Note that we could have equivalently selected $\hat{\mathcal{S}} = \{s|\rho_{\pi_e}(s) > 0\}$ since $\rho_{\pi_e}(s) > 0$ if and only $h(s) > 0$.)

---

**Algorithm 1** Resetting based on demonstrator trajectories

---
1: **procedure** LEARNEDESTOP($M, \mathfrak{A}, \pi_e, n, \xi$)
2:     Rollout multiple trajectories from $\pi_e$: $D \leftarrow [s_1^{(1)}, \ldots, s_H^{(1)}], \ldots, [s_1^{(n)}, \ldots, s_H^{(n)}]$
3:     Estimate the hitting probabilities: $\hat{h}(s) = \frac{1}{n} \sum_i \mathbb{I}\{s \in \tau^{(i)}\}$. (Or $\hat{\rho}$ in continuous settings.)
4:     Construct the smallest $\hat{S}$ allowed by the $\sum_{s \in S \backslash \hat{S}} \hat{h}(s) \le \xi$ constraint.
5:     Add e-stops, resulting in a modified MDP, $\widehat{M}$, where $P_{\hat{S}}(s, a, s'), R_{\hat{S}}(s, a, s') \leftarrow Eq.$ (4)
6: **return** $\mathfrak{A}(\widehat{M})$

---

### 4.3 Imperfect e-stops

In a more realistic setting, consider what happens when we "remove" (i.e. $s \notin \hat{S}$) states as e-stops that have low but non-zero probability of visitation under $\pi_e$. This can happen by "accident" if the supervisor interventions are noisy or we incorrectly estimate the visitation probability to be zero. Alternatively, this can be done intentionally to trade off asymptotic performance for better sample complexity, in which case we remove states with known low but non-zero visitation probability.

**Theorem 4.2.** *Consider $\widehat{M}$, an e-stop variation on MDP $M$ with state spaces $\hat{S}$ and $S$, respectively. Given an expert policy, $\pi_e$, let $h(s)$ denote the probability of visiting state $s$ at least once in an episode roll-out of policy $\pi_e$ in $M$. Then*

$$J(\pi_e) - J(\hat{\pi}^*) \le H \sum_{s \in S \backslash \hat{S}} h(s) \tag{6}$$

*where $\hat{\pi}^*$ is the optimal policy in $\widehat{M}$. Naturally if we satisfy some "allowance," $\xi$, such that $\sum_{s \in S \backslash \hat{S}} h(s) \le \xi$ then $J(\pi_e) - J(\hat{\pi}^*) \le \xi H$.*

**Corollary 4.2.1.** *Recall that $\rho_{\pi_e}(s)$ denotes the average state distribution following actions from $\pi_e$, $\rho_{\pi_e}(s) = \frac{1}{H} \sum_{t=0}^{H-1} \rho_{\pi_e}^t(s)$. Then*

$$J(\pi_e) - J(\hat{\pi}^*) \le \rho_{\pi_e}(S \backslash \hat{S}) H^2 \tag{7}$$

In other words, removing states with non-zero hitting probability introduces error into the policy $\hat{\pi}^*$ according to the visitation probabilities $h$.

**Remark.** The primary slack in these bounds is due to upper bounding the expected cumulative reward for a given state trajectory by $H$. Although this bound is necessary in the worst case, it's worth noting that performance is much stronger in practice. In non-adversarial settings the expected cumulative reward of a state sequence, $\tau$, is correlated with the visitation probabilities of the states along its path: very low reward trajectories tend to have low visitation probabilities, assuming sensible expert policies. We opted against making any assumptions about the correlation between $h(s)$ and the value function, $V(s)$, so this remains an interesting option for future work.

## 5 Learning from observation

In the previous section, we considered how to incorporate general e-stop interventions – which could take the form of an expert supervisor or some other learned e-stop device. Here, we propose and analyze a method for building such a learned e-stop trigger using state observations from an expert demonstrator. This is especially relevant for domains where action observations are unavailable (e.g. videos).

Consider the setting where we observe $n$ roll-outs $\tau^{(1)}, \ldots, \tau^{(n)}$ of a demonstrator policy $\pi_e$ in $M$. We can estimate the hitting probability $h(s)$ empirically as $\hat{h}(s) = \frac{1}{n} \sum_i \mathbb{I}\{s \in \tau^{(i)}\}$. Next, Theorem 4.2 suggests constructing $\hat{S}$ by removing states from $S$ with the lowest $\hat{h}(s)$ values as long as is allowed by the $\sum_{s \in S \backslash \hat{S}} \hat{h}(s) \le \xi$ constraint. In other words, we should attempt to remove as many states as possible while considering our "budget" $\xi$. The algorithm is summarized in Algorithm 1. In practice, implementing Algorithm 1 is actually even simpler: pick $\hat{S}$, take any off-the-shelf implementation and simply end training roll-outs whenever the state leaves $\hat{S}$.

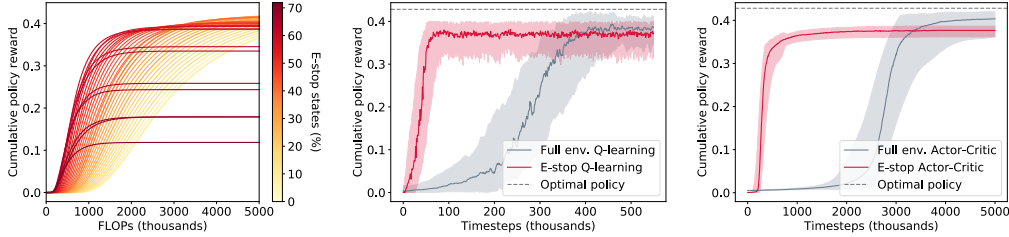

Figure 2: *Left:* Value iteration results with varying portions of the state space replaced with e-stops. Color denotes the portion of states that have been replaced. Note that significant performance improvements may be realized before the optimal policy reward is meaningfully affected. *Middle:* Q-learning results with and without the e-stop mechanism. *Right:* Actor-critic results with and without the e-stop mechanism. Both plots show results across 100 trials. We observe that e-stopping produces drastic improvements in sample efficiency while introducing only a small sub-optimality gap.

**Theorem 5.1.** *The e-stop MDP $\widehat{M}$ with states $\hat{\mathcal{S}}$ in Algorithm 1 has asymptotic sub-optimality*

$$J(\pi_e) - J(\hat{\pi}^*) \le (\xi + \epsilon)H \tag{8}$$

*with probability at least $1 - |\mathcal{S}|e^{-2\epsilon^2 n/|\mathcal{S}|^2}$, for any $\epsilon > 0$. Here $\xi$ denotes our approximate state removal "allowance", where we satisfy $\sum_{s \in \mathcal{S} \setminus \hat{\mathcal{S}}} \hat{h}(s) \le \xi$ in our construction of $\widehat{M}$ as in Theorem 4.2.*

As expected, there exists a tradeoff between the number of trajectories collected, $n$, the state removal allowance, $\xi$, and the asymptotic sub-optimality gap, $J(\pi_e) - J(\hat{\pi}^*)$. In practice we find performance to be fairly robust to $n$, as well as the quality of the expert policy. See Section 6.1 for experimental results measuring the impact of each of these variables.

Note that although this analysis only applies to the discrete setting, the same method can be extended to the continuous case by estimating and thresholding on $\rho_{\pi_e}(s)$ in place of $h(s)$, as implied by Corollary 4.2.1. In Section 6.2 we provide empirical results in continuous domains.

# 6 Empirical study

## 6.1 Discrete environments

We evaluate LEARNEDESTOP on a modified FrozenLake-v0 environment from the OpenAI gym. This environment is highly stochastic: for example, taking a left action can move the character either up, left, or down each with probability $1/3$. To illustrate our ability to evade states that are low-value but non-terminating, we additionally allow the agent to "escape" the holes in the map and follow the usual dynamics with probability 0.01. As in the original problem, the goal state is terminal and the agent receives a reward of 1 upon reaching the goal and 0 elsewhere. To encourage the agent to reach the goal quickly, we use a discount factor of $\gamma = 0.99$.

Across all of our experiments, we observe that algorithms modified with our e-stop mechanism are far more sample efficient thanks to our ability to abandon episodes that do not match the behavior of the expert. We witnessed these benefits across both planning and reinforcement learning algorithms, and with both tabular and policy gradient-based techniques.

Although replacing states with e-stops introduces a small sub-optimality gap, practical users need not despair: any policy trained in a constrained e-stop environment is portable to the full environment. Therefore using e-stops to warm-start learning on the full environment may provide a "best of both worlds" scenario. Annealing this process could also have a comparable effect.

**Value iteration.** To elucidate the relationship between the size of the support set, $\hat{S}$, and the sub-optimality gap, $J(\pi_e) - J(\hat{\pi}^*)$, we run value iteration on e-stop environments with progressively more e-stop states. First, the optimal policy with respect to the full environment is computed and treated as the expert policy, $\pi_e$. Next, we calculate $\rho_{\pi_e}(s)$ for all states. By progressively thresholding

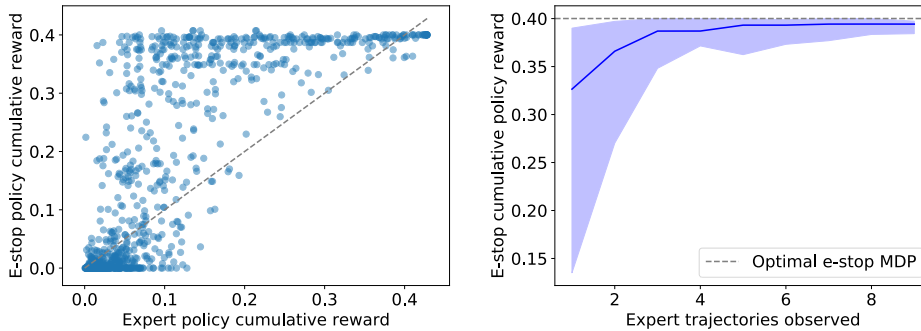

Figure 3: *Left:* E-stop results based on sub-optimal expert policies. *Right:* The number of expert trajectories used to construct $\hat{S}$ vs the final performance in the e-stop environment. E-stop results seem to be quite robust to poor experts and limited demonstrations.

on $\rho_{\pi_e}(s)$ we produce sparser and sparser e-stop variants of the original environment. The results of value iteration run on each of these variants is shown in Fig. 2 (left). Lines are colored according to the portion of states removed from the original environment, darker lines indicating more aggressive pruning. As expected we observe a tradeoff: decreasing the size of $\hat{S}$ introduces sub-optimality but speeds up convergence. Once pruning becomes too aggressive we see that it begins to remove states crucial to reaching the goal and $J(\pi_e) - J(\hat{\pi}^*)$ is more severely impacted as a result.

**RL results.** To evaluate the potential of e-stops for accelerating reinforcement learning methods we ran LEARNEDESTOP from Algorithm 1 with the optimal policy as $\pi_e$. Half of the states with the lowest hitting probabilities were replaced with e-stops. Finally, we ran classic RL algorithms on the resulting e-stop MDP. Fig. 2 (middle) presents our Q-learning results, demonstrating removing half of the states has a minor effect on asymptotic performance but dramatically improves the convergence rate. We also found the e-stop technique to be an effective means of accelerating policy gradient methods. Fig. 2 (right) presents results using one-step actor-critic with a tabular, value function critic [23]. In both cases, we witnessed drastic speedups with the use of e-stops relative to running on the full environment.

**Expert sub-optimality.** The bounds presented Section 4.3 are all in terms of $J(\pi_e) - J(\hat{\pi}^*)$, prompting the question: To what extent is e-stop performance dependent on the quality of the expert, $J(\pi_e)$? Is it possible to exceed the performance of $\pi_e$ as in Theorem 4.1, even with "imperfect" e-stops? To address these questions we artificially created sub-optimal policies by adding noise to the optimal policy's $Q$-function. Next, we used these sub-optimal policies to construct e-stop MDPs, and calculated $J(\hat{\pi}^*)$. As shown in Fig. 3 (left), e-stop performance is quite robust to the expert quality. Ultimately we only need to take care of capturing a "good enough" set of states in order for the e-stop policy to succeed.

**Estimation error.** The sole source of error in Algorithm 1 comes from the estimation of hitting probabilities via a finite set of $n$ expert roll-outs. Theorem 5.1 suggests that the probability of failure in empirically building an e-stop MDP decays exponentially in terms of the number of roll-outs, $n$. We test the relationship between $n$ and $J(\hat{\pi}^*)$ experimentally in Fig. 3 (right) and find that in this particular case it's possible to construct very good e-stop MDPs with as few as 10 expert roll-outs.

## 6.2 Continuous environments

To experimentally evaluate the power of e-stops in continuous domains we took two classic continuous control problems: inverted pendulum control and the HalfCheetah-v3 environment from the OpenAI gym [7], and evaluated the performance of a deep reinforcement learning algorithm in the original environments as well as in modified versions of the environments with e-stops.

Although e-stops are more amenable to analysis in discrete MDPs, nothing fundamentally limits them from being applied to continuous environments in a principled fashion. The notion of state-hitting probabilities, $h(s)$, is meaningless in continuous spaces but the stationary (infinite-horizon), or average state (finite-horizon) distribution, $\rho_{\pi_e}(s)$ is well-defined and many techniques exist for

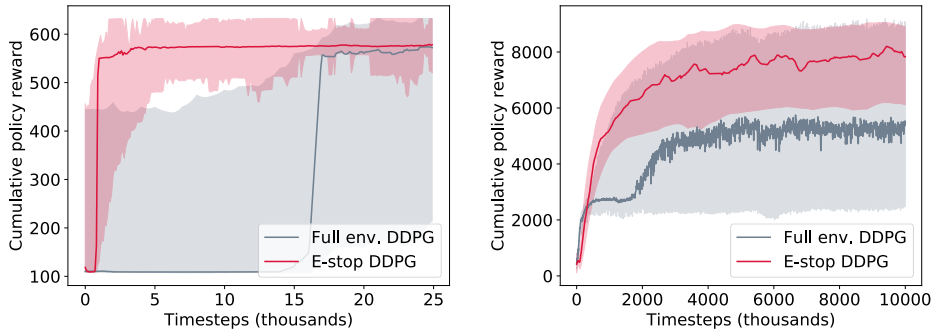

Figure 4: *Left:* DDPG results on the pendulum environment. *Right:* Results on the HalfCheetah-v3 environment from the OpenAI gym. All experiments were repeated with 48 different random seeds. Note that in both cases e-stop agents converged much more quickly and with lower variance than their full environment counterparts.

density estimation in continuous spaces. Applying these techniques along with Corollary 4.2.1 provides fairly solid theoretical grounds for using e-stops in continuous problems.

For the sake of simplicity we implemented e-stops as min/max bounds on state values. For each environment we trained a number of policies in the full environments, measured their performance, and calculated e-stop min/max bounds based on roll-outs of the resulting best policy. We found that even an approach as simple as this can be surprisingly effective in terms of improving sample complexity and stabilizing the learning process.

**Inverted pendulum.** In this environment the agent is tasked with balancing an inverted pendulum from a random starting semi-upright position and velocity. The agent can apply rotational torque to the pendulum to control its movement. We found that without any intervention the agent would initially just spin the pendulum as quickly as possible and would only eventually learn to actually balance the pendulum appropriately. However, the e-stop version of the problem was not tempted with this strange behavior since the agent would quickly learn to keep the rotational velocity to reasonable levels and therefore converged far faster and more reliably as shown in Fig. 4 (left).

**Half cheetah.** Fig. 4 (right) shows results on the HalfCheetah-v3 environment. In this environment, the agent is tasked with controling a 2-dimensional cheetah model to run as quickly as possible. Again, we see that the e-stop agents converged much more quickly and reliably to solutions that were meaningfully superior to DDPG policies trained on the full environment. We found that many policies without e-stop interventions ended up trapped in local minima, e.g. flipping the cheetah over and scooting instead of running. Because e-stops were able to eliminate these states altogether, policies trained in the e-stop regime consistently outperformed policies trained in the standard environment.

Broadly, we found training with e-stops to be far faster and more robust than without. In our experiments, we considered support sets $\hat{S}$ to be axis-aligned boxes in the state space. It stands to reason that further gains could be squeezed out of this framework by estimating $\rho_{\pi_e}(s)$ more prudently and triggering e-stops whenever our estimation, $\hat{\rho}_{\pi_e}(s)$, falls below some threshold. In general, results will certainly be dependent on the structure of the support set used and the parameterization of state space, but our results suggest that there is promise in the tasteful application of e-stops to continuous RL problems.

## 7 Types of support sets and their tradeoffs

In the previous sections, we proposed reset mechanisms based on a continuous or discrete state support set $\hat{S}$. In this section, we describe alternative support set constructions and their respective trade-offs.

At the most basic level, consider the sequence of sets $S_{\pi_e}^1, \ldots, S_{\pi_e}^H$, defined by $S_{\pi_e}^t = \{s | \rho_{\pi_e}^t > \epsilon\}$ for some small $\epsilon$. Note that the single set $\hat{S}$ we considered in Section 4.2 is the union of these sets

when $\epsilon = 0$. The advantage of using a sequence of time-dependent support sets is that $\mathcal{S}_{\pi_e}$ may significantly over-support the expert's state distribution at any time and not reset when it is desirable to do so, i.e. $s_t \in \mathcal{S}_{\pi_e}$ but $s_t \notin \mathcal{S}_{\pi_e}^t$ for some $t > 0$. The downside of using time-dependent sets is that it increases the memory complexity from $\mathcal{O}(|\mathcal{S}|)$ to $\mathcal{O}(|\mathcal{S}|H)$. Further, if the state distributions $\rho_{\pi_e}^1, \ldots, \rho_{\pi_e}^H$ are similar, then using their union effectively increases the number of demonstrations by a factor of $H$.

To illustrate a practical scenario where it is advantageous to use time-dependent sets, we revisit the example in Fig. 1, where an agent navigates from a start state $s_0$. $\mathcal{S}_{\pi_e}$ does not prevent $\pi$ from remaining at $s_0$ for the duration of the episode, as $\pi_e$ is initialized at this state and thus $s_0 \in \mathcal{S}_{\pi_e}$. Clearly, this type of behavior is undesirable, as it does not move the agent towards its goal. However, the time-dependent sets would trigger an intervention after only a couple time steps, since $s_0 \in \mathcal{S}_{\pi_e}^0$ but $s_0 \in \mathcal{S}_{\pi_e}^t$ for some $t > 0$.

Finally, we propose an alternative support set based on visitation counts, which balances the trade-offs of the two previous constructions $\mathcal{S}_{\pi_e}$ and $\{\mathcal{S}_{\pi_e}^1, \ldots, \mathcal{S}_{\pi_e}^H\}$. Let $s_f \in \mathbb{N}_0^{|\mathcal{S}|}$ be an auxiliary state, which denotes the number of visits to each state. Let $f(s) = \sum_{t=1}^{H} \mathbb{I}\{s \in \mathcal{S}_{\pi_e}^t\}$ be the visitation count to state $s$ by the demonstrator. The modified MDP in this setting is defined by

$$
P_{\hat{\mathcal{S}}}(s, s_f, a, s', s_f') = \begin{cases} P(s, a, s'), & \text{if } s_f \leq f(s), s_f' = s_f + e_s \\ 1, & \text{if } s_f > f(s), s' = s_{\text{term}}, s_f' = s_f + e_s \\ 0 & \text{else} \end{cases}
$$

$$
R_{\hat{\mathcal{S}}}(s, s_f, a, s') = \begin{cases} R(s, a, s'), & \text{if } s_f \leq f(s) \\ 0, & \text{else} \end{cases}
$$

(9)

where $e_s$ is the one-hot vector for state $s$. In other words, we terminate the episode with no further reward whenever the agent visits a state more than the demonstrator. The mechanism in Eq. (9) has memory requirements independent of $H$ yet fixes some of the over-support issues in $\mathcal{S}_{\pi_e}$. The optimal policy in this MDP achieves at least as much cumulative reward as $\pi_e$ (by extending Theorem 4.1) and can be extend to the imperfect e-stop setting in Section 4.3.

We leave exploration of these and other potential e-stop constructions to future work.

# 8    Conclusions

We introduced a general framework for incorporating e-stop interventions into any reinforcement learning algorithm, and proposed a method for learning such e-stop triggers from state-only observations. Our key insight is that only a small support set of states may be necessary to operate effectively towards some goal, and we contribute a set of bounds that relate the performance of an agent trained in this smaller support set to the performance of the expert policy. Tuning the size of the support set allows us to efficiently trade off an asymptotic sub-optimality gap for significantly lower sample complexity.

Empirical results on discrete and continuous environments demonstrate significantly faster convergence on a variety of problems and only a small asymptotic sub-optimality gap, if any at all. We argue this trade-off is beneficial in problems where environment interactions are expensive, and we are less concerned with achieving no-regret guarantees as we are with small, finite sample performance. Further, such a trade-off may be beneficial during initial experimentation and for bootstrapping policies in larger state spaces. For example, we are particularly excited about graduated learning processes that could increase the size of the support set over time.

In larger, high dimensional state spaces, it would be interesting and relatively straightforward to apply anomaly detectors such as one-class SVMs [16] or auto-encoders [19] within our framework to implicitly construct the support set. Our bounds capture the reduction in exploration due to reducing the state space size, which could be further tightened by incorporating our a priori knowledge of $s_{\text{term}}$ and the ability to terminate trajectories early.

## 9 Acknowledgements

The authors would like to thank The Notorious B.I.G. and Justin Fu for their contributions to music, pop culture, and our implementation of DDPG. This work was (partially) funded by the National Science Foundation TRIPODS+X:RES (#A135918), National Institute of Health R01 (#R01EB019335), National Science Foundation CPS (#1544797), National Science Foundation NRI (#1637748), the Office of Naval Research, the RCTA, Amazon, and Honda Research Institute USA.

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
