[Supplementary Material]

# Appendix for "Mo′ States Mo′ Problems: Emergency Stop Mechanisms from Observation"

## A Proof of Eq. (5)

**Lemma.** *Let $M = \langle \mathcal{S}, \mathcal{A}, P, R, H, \rho_0 \rangle$ be a finite horizon, episodic Markov decision process, and $\widehat{M} = (\hat{\mathcal{S}}, \mathcal{A}, P_{\hat{\mathcal{S}}}, R_{\hat{\mathcal{S}}}, H, \rho_0)$ be a corresponding e-stop version of $M$. Given a reinforcement learning algorithm $\mathfrak{A}$, the regret in $M$ after running $\mathfrak{A}$ for $T$ timesteps in $\widehat{M}$ is bounded by*

$$\text{Regret}(T) \leq \left\lceil \frac{T}{H} \right\rceil [J(\pi^*) - J(\hat{\pi}^*)] + \mathbb{E}_{\widehat{M}}^{\hat{\pi}^*}[R_T] - \mathbb{E}_{\widehat{M}}^{\mathfrak{A}}[R_T] \tag{10}$$

*where $\pi^*$ and $\hat{\pi}^*$ are the optimal policies in $M$ and $\widehat{M}$, respectively*

*Proof.* Let $\hat{\pi}^* = \arg\max_{\pi \in \Pi} J_{\widehat{M}}(\pi)$. Then beginning with the external regret definition,

$$\text{Regret}_M^{\mathfrak{A}}(T) = \mathbb{E}_M^{\pi_e}[R_T] - \mathbb{E}_M^{\mathfrak{A}}[R_T] \tag{11}$$

$$= \left[ \mathbb{E}_M^{\pi_e}[R_T] - \mathbb{E}_{\widehat{M}}^{\hat{\pi}^*}[R_T] \right] + \left[ \mathbb{E}_{\widehat{M}}^{\hat{\pi}^*}[R_T] - \mathbb{E}_{\widehat{M}}^{\mathfrak{A}}[R_T] \right] + \left[ \mathbb{E}_{\widehat{M}}^{\mathfrak{A}}[R_T] - \mathbb{E}_M^{\mathfrak{A}}[R_T] \right] \tag{12}$$

$$\leq \left[ \mathbb{E}_M^{\pi_e}[R_T] - \mathbb{E}_{\widehat{M}}^{\hat{\pi}^*}[R_T] \right] + \left[ \mathbb{E}_{\widehat{M}}^{\hat{\pi}^*}[R_T] - \mathbb{E}_{\widehat{M}}^{\mathfrak{A}}[R_T] \right] \tag{13}$$

$$\leq \left[ \mathbb{E}_M^{\pi_e}[R_T] - \mathbb{E}_M^{\hat{\pi}^*}[R_T] \right] + \left[ \mathbb{E}_{\widehat{M}}^{\hat{\pi}^*}[R_T] - \mathbb{E}_{\widehat{M}}^{\mathfrak{A}}[R_T] \right] \tag{14}$$

$$\leq \left\lceil \frac{T}{H} \right\rceil [J(\pi_e) - J(\hat{\pi}^*)] + \left[ \mathbb{E}_{\widehat{M}}^{\hat{\pi}^*}[R_T] - \mathbb{E}_{\widehat{M}}^{\mathfrak{A}}[R_T] \right] \tag{15}$$

Eq. (13) follows by the definition of $\widehat{M}$. □

## B Proof of Theorem 4.1

**Theorem.** *Suppose $\widehat{M}$ is an e-stop variant of $M$ such that $\hat{\mathcal{S}} = \{s | h(s) > 0\}$ where $h(s)$ denotes the probability of hitting state $s$ in a roll-out of $\pi_e$. Let $\hat{\pi}^* = \arg\max_{\pi \in \Pi} J_{\widehat{M}}(\pi)$ be the optimal policy in $\widehat{M}$. Then $J(\hat{\pi}^*) \geq J(\pi_e)$.*

*Proof.* Let $J_{\widehat{M}}(\pi)$ denote the value of executing policy $\pi$ in $M_{\hat{\mathcal{S}}}$. By the definition of $M_{\hat{\mathcal{S}}}$,

$$J(\pi) \geq J_{\widehat{M}}(\pi) \quad \forall \pi \tag{16}$$

$$J_{\widehat{M}}(\pi_e) = J(\pi_e) \tag{17}$$

because $\pi_e$ never leaves $\mathcal{S}_{\pi_e}$ (and thus never leaves $\hat{\mathcal{S}}$). Finally, by the definition of $\hat{\pi}^*$ and the realizability of $\pi_e$,

$$J_{\widehat{M}}(\hat{\pi}^*) \geq J_{\widehat{M}}(\pi_e) \tag{18}$$

Combining Eqs. (16) to (18) implies $J(\hat{\pi}^*) > J(\pi_e)$. □

## C Proof of Theorem 4.2

**Theorem.** *Consider $\widehat{M}$, an e-stop variation on MDP $M$ with state spaces $\hat{\mathcal{S}}$ and $\mathcal{S}$, respectively. Given an expert policy, $\pi_e$, let $h(s)$ denote the probability of visiting state $s$ at least once in an episode roll-out of policy $\pi_e$ in $M$. Then*

$$J(\pi_e) - J(\hat{\pi}^*) \leq H \sum_{s \in \mathcal{S} \setminus \hat{\mathcal{S}}} h(s) \tag{19}$$

*where $\hat{\pi}^*$ is the optimal policy in $\widehat{M}$. Naturally if we satisfy some "allowance," $\xi$, such that $\sum_{s \in \mathcal{S} \setminus \hat{\mathcal{S}}} h(s) \leq \xi$ then $J(\pi_e) - J(\hat{\pi}^*) \leq \xi H$.*

*Proof.* We proceed by analyzing the probabilities and expected rewards of entire trajectories $\tau = (\tau_1, \ldots, \tau_H)$, in $M$ and $\widehat{M}$. Let

$$\mu(\tau) = \sum_{t=1}^{H-1} \mathbb{E}\left[R(\tau_t, A_t, \tau_{t+1}) | \tau, \pi_e\right] \tag{20}$$

be the expected reward of a trajectory $\tau$ and let $p_M(\tau)$ denote the probability of trajectory $\tau$ when following policy $\pi_e$ in MDP $M$. Note that

$$h(s) = \sum_\tau p_S(\tau)\mathbb{I}\{s \in \tau\} \tag{21}$$

Now,

$$J(\pi_e) - J(\hat{\pi}^*) \leq J_M(\pi_e) - J_{\widehat{M}}(\hat{\pi}^*) \tag{22}$$

$$\leq J_M(\pi_e) - J_{\widehat{M}}(\pi_e) \tag{23}$$

$$= \sum_\tau p_S(\tau)\mu(\tau) - \sum_\tau p_{\hat{S}}(\tau)\mu(\tau) \tag{24}$$

$$\leq \sum_\tau p_S(\tau)\mu(\tau)\mathbb{I}\{\tau \text{ leaves } \hat{S}\} \tag{25}$$

$$\leq H \sum_\tau p_S(\tau)\mathbb{I}\{\tau \text{ leaves } \hat{S}\} \tag{26}$$

$$\leq H \sum_\tau p_S(\tau) \sum_{s \in S \setminus \hat{S}} \mathbb{I}\{s \in \tau\} \tag{27}$$

$$= H \sum_{s \in S \setminus \hat{S}} \sum_\tau p_S(\tau)\mathbb{I}\{s \in \tau\} \tag{28}$$

$$= H \sum_{s \in S \setminus \hat{S}} h(s) \tag{29}$$

as desired. $\qquad\square$

# D   Proof of Corollary 4.2.1

**Corollary.** *Recall that $\rho_{\pi_e}(s)$ denotes the average state distribution following actions from $\pi_e$, $\rho_{\pi_e}(s) = \frac{1}{H}\sum_{t=0}^{H-1}\rho_{\pi_e}^t(s)$. Then*

$$J(\pi_e) - J(\hat{\pi}^*) \leq \rho_{\pi_e}(\mathcal{S} \setminus \hat{\mathcal{S}})H^2 \tag{30}$$

*Proof.* Note that

$$h(s) = \mathbb{P}\left(\bigcup_{t=0}^{H-1}(s_t = s)\right) \leq \sum_{t=0}^{H-1}\rho_{\pi_e}^t(s) = H\rho_{\pi_e}(s) \tag{31}$$

where the inequality follows from a union bound over time steps. Then

$$J(\pi_e) - J(\hat{\pi}^*) \leq \rho_{\pi_e}(S \setminus \hat{S})H^2 \tag{32}$$

as a consequence of Theorem 4.2. $\qquad\square$

# E   Proof of Theorem 5.1

**Theorem.** *The e-stop MDP $\widehat{M}$ with states $\hat{S}$ in Algorithm 1 has asymptotic sub-optimality*

$$J(\pi_e) - J(\hat{\pi}^*) \leq (\xi + \epsilon)H \tag{33}$$

*with probability at least $1 - |\mathcal{S}|e^{-2\epsilon^2 n/|\mathcal{S}|^2}$, for any $\epsilon > 0$. Here $\xi$ denotes our approximate state removal "allowance", where we satisfy $\sum_{s \in \mathcal{S} \setminus \hat{\mathcal{S}}} \hat{h}(s) \leq \xi$ in our construction of $\widehat{M}$ as in Theorem 4.2.*

*Proof.* With Hoeffding's inequality and a union bound,

$$\mathbb{P}(\forall s, \hat{h}(s) > h(s) - \epsilon/|\mathcal{S}|) = 1 - \mathbb{P}(\exists s, \hat{h}(s) \leq h(s) - \epsilon/|\mathcal{S}|) \tag{34}$$

$$\geq 1 - |\mathcal{S}|e^{-2\epsilon^2 n/|\mathcal{S}|^2} \tag{35}$$

Note that the $\hat{h}(s)$ values are not independent yet the union bound still allows us to bound the probability that any of them deviate meaningfully from $h(s)$. Now if $\hat{h}(s) > h(s) - \epsilon/|\mathcal{S}|$ for all $s$, it follows that

$$\xi \geq \sum_{s \in \mathcal{S} \setminus \hat{\mathcal{S}}} h(s) - \frac{\epsilon}{|\mathcal{S}|}(|\mathcal{S}| - |\hat{\mathcal{S}}|) \geq \sum_{s \in \mathcal{S} \setminus \hat{\mathcal{S}}} h(s) - \epsilon \tag{36}$$

and so $\sum_{s \in \mathcal{S} \setminus \hat{\mathcal{S}}} h(s) \leq \xi + \epsilon$. By Theorem 4.2 we have that

$$J(\pi_e) - J(\hat{\pi}^*) \leq (\xi + \epsilon)H \tag{37}$$

completing the proof. □

# F   Imperfect e-stops in terms of $\hat{\rho}_{\pi_e}(s)$

While Theorem 5.1 provides an analysis for an approximate e-stopping algorithm, its reliance on hitting probabilities does not extend nicely to continuous domains. Here we present a result analogous to Theorem 5.1, but using $\hat{\rho}_{\pi_e}(s)$ in place of $\hat{h}(s)$. Unfortunately, we are not able to escape a dependence on $|\mathcal{S}|$ with this approach however. Furthermore, we require that $\pi_e$ always runs to episode completion without hitting any terminal states, ie. the length of all $\pi_e$ roll-outs is $H$.

**Definition F.1.** Let $\varrho(s)$ be a random variable denoting the average number of times $\pi_e$ visits state $s$,

$$\varrho(s) \triangleq \frac{|\{t \in 1, \ldots, H | \tau_t = s\}|}{H}. \tag{38}$$

Note that with $n$ roll-outs, our approximate average state distribution is the same as the average of the $\varrho$'s:

$$\hat{\rho}_{\pi_e}(s) \triangleq \frac{1}{nH} \sum_{i=1}^{n} \sum_{t=1}^{H} \mathbb{I}\{\tau_t^{(i)} = s\} = \frac{1}{n} \sum_{i=1}^{n} \varrho^{(i)}(s).$$

**Theorem F.1.** *The e-stop MDP $\widehat{M}$ with states $\hat{\mathcal{S}}$ resulting from running the $\hat{\rho}_{\pi_e}$ version of Algorithm 1 with $n$ expert roll-outs has asymptotic sub-optimality*

$$J(\pi_e) - J(\hat{\pi}^*) \leq \left( \xi + \sqrt{\frac{2 \log(2|\mathcal{S}|/\delta)}{n}} \sum_{s \in \mathcal{S}} \sqrt{V_n(\varrho^{(1:n)}(s))} + \frac{7|\mathcal{S}| \log(2|\mathcal{S}|/\delta)}{3(n-1)} \right) H^2 \tag{39}$$

*with probability at least $1 - \delta$. Here $\xi$ denotes our approximate state removal "allowance", where we satisfy $\hat{\rho}_{\pi_e}(\mathcal{S} \setminus \hat{\mathcal{S}}) \leq \xi$ in our construction of $\widehat{M}$, and $V_n$ denotes the sample variance.*

*Proof.* We follow the same structure as in the proof of Theorem 5.1, but use an empirical Bernstein bound in place of Hoeffding's inequality. We know from Theorem 4 of Maurer and Pontil [18] that, for each $s$,

$$\hat{\rho}_{\pi_e}(s) \leq \rho_{\pi_e}(s) - \left( \sqrt{\frac{2V_n(\varrho^{(1:n)}(s)) \log(2|\mathcal{S}|/\delta)}{n}} + \frac{7 \log(2|\mathcal{S}|/\delta)}{3(n-1)} \right) \tag{40}$$

with probability no more than $\delta/|\mathcal{S}|$. It follows that it will hold for *every* $s \in \mathcal{S}$ with probability at least $1 - \delta$. In that case we underestimated the true $\rho$-mass of any subset of the state space $\mathcal{S}$ by at most

$$\sqrt{\frac{2 \log(2|\mathcal{S}|/\delta)}{n}} \sum_{s \in \mathcal{S}} \sqrt{V_n(\varrho^{(1:n)}(s))} + \frac{7|\mathcal{S}| \log(2|\mathcal{S}|/\delta)}{3(n-1)} \tag{41}$$

and so by Theorem 4.2 we have the desired result. □

# G  Experimental details

All experiments were implemented with Numpy and JAX [6]. NuvemFS (https://nuvemfs.com) was used to manage code and experimental results. Experiments were run on AWS.

Our code and results are available on GitHub at https://github.com/samuela/e-stops.

Figure 5: Our FrozenLake-v0 environment. The agent starts in the upper left square and attempts to reach the goal in the lower right square. Tiles marked with "H" are holes in the lake which the agent can fall in and recover with only probability 0.01. The optimal state-value function is overlaid.

## G.1  Value iteration

We ran value iteration on the full environment to convergence (tolerance $1e{-}6$) to establish the optimal policy. We calculated the state hitting probabilities of this policy exactly through an interpretation of expert policy roll-outs as absorbing Markov chains. These hitting probabilities were then ranked and states were removed in order of their rank until there was no longer a feasible path to the goal ($J(\pi) = 0$). The number of floating point operations (FLOPs) used was calculated based on $4\,|\mathcal{S}|^2|A|$ FLOPs per value iteration update:

1. For each state $s$ and each action $a$, calculating the expected value of the next state. ($|\mathcal{S}||A|$ dot products of $|\mathcal{S}|$-vectors.)

2. Multiplying those values by $\gamma$.

3. Adding in the expected rewards for every state-action-state transition.

4. Calculating the maximum for each state $s$ and each action $a$ over $|\mathcal{S}|$ possible next state outcomes.

## G.2  Policy gradient methods

We ran value iteration on the full environment to convergence (tolerance $1e - 6$) to establish the optimal policy. We estimated the state hitting probabilities of this policy with 1,000 roll-outs in the environment. Based on this estimate of $\rho_{\pi_e}(s)$ we replaced the least-visited 50% of states with e-stops.

We ran both Q-learning and Actor-Critic across 96 trials (random seeds 0-95) and plot the median performance per states seen. Error bars denote one standard deviation around the mean and are clipped to the maximum/minimum values. We ran iterative policy evaluation to convergence on the current policy every 10 episodes in order to calculate the cumulative policy reward as plotted.

In order to accommodate the fact that two trials may not have x-coordinates that align (episodes may not be the same length), we linearly interpolated values and plot every 1,000 states seen.

### G.3 DDPG

Continuous results were trained with DDPG with $\gamma = 0.99, \tau = 0.0001$, Adam with learning rate 0.001, batch size 128, and action noise that was normally distributed with mean zero and standard deviation 0.1. The replay buffer had length $2^{20} = 1,048,576$. The actor network had structure

- Dense(64)
- ReLU
- Dense(64)
- ReLU
- Dense(action_shape)
- Tanh

and the critic network had structure

- Dense(64)
- ReLU
- Dense(64)
- ReLU
- Dense(64)
- ReLU
- Dense(1)

We periodically paused training to run policy evaluation on the current policy (without any action noise).

Plotting and error bars are the same as in the deterministic experiments.