[Reviews · NeurIPS 2019]

Reviewer 1



The paper proposes a method for improving convergence rates of RL algorithms when one has access to a set of state-only expert demonstrations. The method works by modifying the given MDP so that the episode terminates whenever the agent leaves the set of states that had high-probability under the expert demonstrations. The paper then proves an upper bound on the regret incurred using their algorithm (as compared to the expert) in terms of the regret for the RL algorithm that is used to solve the modified MDP. The paper presents a set of experiments showing that the proposed mechanism can effectively strike a tradeoff between convergence rate and optimality. The clarity of the exposition is quite high, and the paper is easy to follow. The idea behind the algorithm is not particularly novel. The theoretical results presented in the paper seem significant in that they lend theoretical support to an idea that makes good intuitive sense, however I am not sufficiently familiar with theoretical work in this area to properly judge the quality and significance of the proved results. One quibble I might state is that nothing is proved about the main motivation behind the algorithm, i.e. improved convergence rates. But maybe such proofs are difficult to pull off. The reported experiments are simple and intuitively appealing, however there is much more that could have been done to characterize the proposed method; the most notable omission is the extension to non-tabular environments. Other comments: I do not understand Definition 5.1. It seems to be taking the argmin of a boolean-valued function. Is the argmin necessary? Quibble: Line 36: "terminate the episode with a large penalty when the agent leaves this set" is a bit misleading since in the experiments and theorems below that agents are just given 0 reward when they leave the support supersets. The recent paper "Off-Policy Deep Reinforcement Learning without Exploration" may be of interest, as it explores similar ideas of constraining policies to the regions explored by the expert, but under somewhat different assumptions. Another quibble I have (this is maybe more of a personal thing, and as such won't affect my score) is that the title is not very descriptive of the work. First, the "in Training" feels unnecessary; surely the majority of improvements in machine learning are about improving training methods. That leaves us with "Fast Agent Resetting". But isn't it true that the point of the work is not that the proposed resetting procedure is particularly fast, but rather that the proposed resetting policy makes *convergence* fast? Thus if I could title the work myself it would be some refined version of "Improving Convergence Rates in Reinforcement Learning Through Adaptive Resetting". This could also be improved to indicate that the resetting is based on expert demonstrations, i.e. to signal to potential readers that the work relates to Learning from Demonstrations. The experiments that were shown are tidy and nicely interpretable, but do not go very far in characterizing the proposed method. Indeed a single tabular environment was tested. It would be a big improvement to see this method applied in any kind of non-tabular environment, and this doesn't seem like such a big ask given that the paper describes this as "straightforward" in the final paragraph. Another thing I would like to see tested is how performance scales as one changes the number of expert trajectories provided; in the Q-learning / actor-critic examples, 1000 expert trajectories are provided, which seems like a lot compared to the complexity of the environment. Typos: Line 52 "general falls" Line 72 should be "model predictive control" rather than "model-predictive control" Line 163 should be \rho_{\pi_d} instead of \rho_{\pi_e} Line 164 missing bar on absolute value for \hat{S}

Reviewer 2



SUMMARY: The authors study the problem of reinforcement learning (RL) when the learning agent has access to state observations from a demonstrator in additional to an environment reward signal. More specifically, the authors wish to leverage the demonstrator-provided state observations to improve the learning efficiency of the learning agent. To do so, the authors propose the use of "e-stops" in order to guide the exploration of the learning agent. STRENGTHS: * The authors propose an attractive and simple way to leverage state-only demonstrations in combination with RL. * The authors provide extensive theoretical analysis of their algorithm. * The proposed method produces convincing empirical results, albeit in a single, simple domain. WEAKNESSES: (A) The authors miss some related work that needs to be commented on for the reader to better understand the contribution: (1) The concept of not straying too far from the demonstrator's behavior sounds similar in spirit to, say, DQfD [R1], where there is a large cost associated with deviating from what the expert would do. Is there a straightforward modification to that algorithm that would apply to the setting considered here? (2) The concept of e-stopping sounds similar to the recently-proposed notion of "shielding" [R2], where one might provide a set of "safe states". How is the proposed method different? [R1] Hester et al. "Deep Q-learning from demonstrations." AAAI. 2018. [R2] Alshiekh et al. "Safe reinforcement learning via shielding." AAAI. 2018. (B) Experiments in more domains would have been more convincing, as would providing any (even exceedingly-simple) baseline for Figure 3 that uses the state-only demonstrations in addition to the environment reward. MINOR COMMENTS: * The authors should define the term "e-stop" for the uninitiated reader.

Reviewer 3



Post Rebuttal: The rebuttal did a nice job of changing my mind. In particular, I was glad to see the method work in continuous spaces and with sub-optimal demonstrations. ------------ This is a well-written paper that is reasonable clear. It introduces the novel ideas of e-stops, which reduces the exploration required, and that such e-stops can be learned from a sequence of observations. The core idea of the paper is that when the agent goes off the rails and is in a bad part of the state space, it should be reset, rather than spending time/samples exploring this area. The main drawbacks of the paper are * the method only works in discrete MDPs * it's not clear how well the method works relative to other imitation learning methods * it's not clear how well the method will work when the demonstrated trajectories are poor "We argue that such interventions can... (b) address function approximation convergence issues introduced by covariate shift.": Is this shown? The results are only in discrete domains and I don't think function approximation is used at all. "We emphasize that our technique for incorporating observations applies to any reinforcement learning algorithm.": This statement seemed to over claim - doesn't it only work on discrete MDPs? It wasn't clear to me why the method introduced in [5] shouldn't be empirically compared to the proposed method. More generally, there are many methods that combine imitation learning from state-only demonstrations with RL - why are these methods not relevant as empirical baselines? There is a general paragraph at the end of section 2 saying why some existing methods cannot be applied - is the argument that *none* of the existing methods apply in the paper's empirical setting? Section 4 makes thet claim that "This modified MDP M_\hat{S} [defined in equation (5)] has an optimal policy which achieves at least the same reward as \pi_d on the true MDP M". Doesn't this depend on the quality of the support superset? For instance, if the support superset guides the agent to a sub-optimal terminal state, and does not allow the agent to reach the optimal terminal state, won't the agent in M_\hat{S} be prohibited from finding the optimal terminal state? In the empirical study, I was expecting to see an experiment where the demonstrations were suboptimal (e.g., travels left and right on the upper edge of the environment). Will the agent still be able to learn a near-optimal policy in this case? Figure 2 right: What is the x-axis? Is it really floating point operations? If so, why is this a reasonable metric? I like that the authors tested their method with value iteration, Q-learning, and actor-critic methods.

[Author Response · NeurIPS 2019]

We thank the reviewers for their thoughtful feedback and suggested improvements. We agree that our proposed e-stop framework is a simple and attractive mechanism for incorporating state-only observations from an expert demonstrator, or manual interventions from a supervisor. Here, we provide clarifications and additional results which address the reviewer concerns necessary to further increase their scores.

**Additional empirical analysis** The central feedback from all reviewers was the need for experiments in a continuous MDP, in addition to our previously presented tabular results. We present new results on an inverted pendulum environment with continuous states and actions which will be included in the manuscript. We trained an agent using deep deterministic policy gradient (DDPG), with an actor and critic using two and three hidden layers of size 64, respectively. We collected 500 demonstration trajectories from a near-optimal expert policy and constructed a support superset from a rectangular hull of these samples. Fig. 1 compares DDPG with and without an e-stop mechanism, and shows roughly an order of magnitude improvement in sample complexity on a hold-out set of random seeds.

We provide further fine-grained analysis by controlling for several other parameters in the grid-world environment. To address [R1], we present results in Fig. 2 over a range of expert demonstrations used to construct the support superset $\hat{S}$. Even after decreasing the number of demonstrations from 1000 to 5, the cumulative reward of the agent trained via our e-stop mechanism decreased by less than 3%. Note that a bound relating the number of expert trajectories to the suboptimality of our method was proved in Theorem 5.1. To address [R3], in Fig. 3, we controlled for the quality of the demonstrator policy by injecting random noise into the optimal policy's $Q$-function. As predicted by Theorem 5.1, the learned e-stop policy has bounded regret with respect to the expert demonstrations, but in practice typically exceeded the performance of the expert.

Figure 1: Pendulum environment.  Figure 2: Number of observations.  Figure 3: Expert quality.

**Relationship to existing work** We emphasize that our method applies to the reinforcement learning with expert observations (RLEO) setting, where only the states visited by the expert are observed. This precludes methods for the reinforcement learning with expert demonstrations (RLED) setting, where both the expert's states and actions are provided. To answer [R2], it is not immediately clear how DQfD (Hester et al.) could be extended to the RLEO setting, as their method requires action observations to perform Bellman updates. However, it is possible to extend our method to RLED by constructing a support superset $\hat{S}$ based on state-action pairs. Thus, our method and many RLED methods are complimentary. For example, DQfD would allow pre-training the policy from the state-action demonstrations, whereas ours reduces exploration during the on-policy learning phase. Similarly, our work can be related to Fujimoto et al. (as suggested by [R1]) by using a state-action superset and off-policy training, where states outside of $\hat{S}$ are never selected in the Bellman update rule due to the maximum terminal e-stop penalty.

To answer [R3], the method in Eisenbach et al. is also complimentary to our work, and does not permit a direct comparison. To elaborate, they consider the RL setting where a second "soft reset" policy is learned in addition to the standard hard reset. The soft reset policy prevents the agent from entering nearly non-reversible states and returns the agent to an initial state. Hard resets are required whenever the soft reset policy fails to terminate in a manually defined set of safe states $\mathcal{S}_{\mathrm{reset}}$. Our method can be seen as learning $\mathcal{S}_{\mathrm{reset}}$ from observation (in the absence of a soft reset policy, we also use $\mathcal{S}_{\mathrm{reset}}$ during the forward policy roll-outs). Their method trades off hard resets for soft resets, whereas ours learns when to perform the hard resets.

"Shielding" (Alshiekh et al.), as suggested by [R2], uses a manually specified safety constraint and a course, conservative abstraction of the dynamics to prevent an agent from violating the safety constraint. Our expert observations can be seen as inducing a safe region (the support superset), although our e-stop mechanism is model-free and uses a terminal reset penalty for actions which would have exited the safe region.

**Other minor improvements** We will make all typographical and clarity changes suggested by the reviewers, including using a more informative title.

We believe these changes, and in particular the additional results in continuous MDPs, address all reviewer concerns.

[Meta-Review · NeurIPS 2019]

The paper proposes a method for stopping unnecessary exploration in RL with a bounded regret on the loss. The stopping method, called e-stop, learns from state-only demonstrations provided by an expert. The paper is very well-written and clear to follow. The theoretical analysis of the method is compelling. The experiments are rather minimalistic, but they support the theoretical analysis. A drawback of this work is that it is limited to discrete MDPs, which could seriously hinder its impact given that modern RL methods are all about continuous high-dimensional state spaces.